# Fingerprinting and chemotyping approaches reveal a wide genetic and metabolic diversity among wild hops (*Humulus lupulus* L.)

Florent Ducrocq[1], Séverine Piutti[1], Alena Henychová[2], Jean Villerd[1], Alexandre Laflotte[3], Loïc Girardeau[1], Jérémy Grosjean[1], Josef Patzak[2], Alain Hehn[1]*

**1** Université de Lorraine, INRAE, LAE, Nancy, France, **2** Hop Research Institute Co. Ltd., Žatec, Czech Republic, **3** Université de Lorraine, Centre de Recherche et Développement de la Bouzule, Nancy, France

* alain.hehn@univ-lorraine.fr

## Abstract

Hop (*Humulus lupulus* L.) is an emblematic industrial crop in the French North East region that developed at the same time as the brewing activity. Presently, this sector, especially microbreweries, are interested in endemic wild hops, which give beer production a local signature. In this study, we investigated the genetic and metabolic diversity of thirty-six wild hops sampled in various ecological environments. These wild accessions were propagated aeroponically and cultivated under uniform conditions (the same soil and the same environmental factors). Our phytochemical approach based on UHPLC-ESI-MS/MS analysis led to the identification of three metabolic clusters based on leaf content and characterized by variations in the contents of twelve specialized metabolites that were identified (including xanthohumol, bitter acids, and their oxidized derivatives). Furthermore, molecular characterization was carried out using sixteen EST-SSR microsatellites, allowing a genetic affiliation of our wild hops with hop varieties cultivated worldwide and wild hops genotyped to date using this method. Genetic proximity was observed for both European wild and hop varieties, especially for Strisselspalt, the historical variety of our region. Finally, our findings collectively assessed the impact of the hop genotype on the chemical phenotype through multivariate regression tree (MRT) analysis. Our results highlighted the 'WRKY 224' allele as a key discriminator between high- and low-metabolite producers. Moreover, the model based on genetic information explained 40% of the variance in the metabolic data. However, despite this strong association, the model lacked predictive power, suggesting that its applicability may be confined to the datasets analyzed.

## Introduction

Hop (*Humulus lupulus* L.) is a dioecious, climbing, and perennial plant belonging to the *Cannabaceae* family [1], widely distributed throughout the Northern Hemisphere

**Data availability statement:** All relevant data are within the article and its Supporting Information files.

**Funding:** The author(s) received no specific funding for this work.

**Competing interests:** The authors have declared that no competing interests exist.

(Europe, Asia, and North America) between latitudes 35° and 55°, even if it is also present in Australia, New Zealand, South Africa, Argentina, and Brazil [2].

*H. lupulus* has been commonly used in the past, especially in traditional medicine as a healing agent [3], and recent studies have increased its applications in the cosmetic, pharmaceutical, and agronomic fields [4]. Today, this plant is cultivated mostly for its female inflorescences, named cones, which are used for aromatic and/or bitter properties in the brewing industry. The flavor and bitterness of beer, as well as the healing properties of this plant, are due to the production of specialized metabolites such as bitter acids, prenylated chalcones (polyphenols), or essential oils containing volatile organic compounds (mostly monoterpenes and sesquiterpenes) [5,6].

These molecules are produced in lupulin glands, which are mostly observed on cones and, to a lesser extent, on the underside of leaves [7]. Several factors have been reported to modulate their biosynthesis, such as hop genotype [8], or environmental factors such as abiotic factors (temperature, radiation, water availability, etc.) [9,10] and biotic factors (viruses, phytopathogens, etc.) [11,12], as well as, more broadly, the interaction between both. To avoid chemical heterogeneity in hop cultivation and meet the criteria of bitterness, aroma, yield, quality, etc., for the brewing industry, wild hops have been selected and domesticated to create hop cultivars. However, this selection process has led to plants that are less tolerant to emerging biotic and abiotic stresses. Many studies highlight the damage caused by various pests, such as downy mildew [13] or powdery mildew [14], to hop cultivars, which can cause severe losses.

Reintroducing wild hops in breeding programs could increase biotic and abiotic tolerance and create new cultivars capable of overcoming climate change [15], as recent studies on wild hops have investigated their potential for future breeding programs [16–27]. Various approaches have been developed to differentiate hops, especially for identifying varieties, on the basis of their metabolic content, such as that of bitter acids [8] or volatile organic compounds [28]. However, these methods are difficult to reproduce since the biosynthesis of these compounds is highly susceptible to environmental changes. To address this limitation, DNA-based molecular techniques have been developed as robust tools for hop genotype evaluation. Initial approaches included RAPD (Random Amplified Polymorphic DNA) [29], and AFLP (Amplify Fragment Length Polymorphism) [30]. However, these methods exhibited low polymorphim levels, and were labor-intensive and costly [30]. Consequently, ISSR (Inter Simple Sequence Repeated) markers were introduced, offering improved reproducibility, and higher polymorphism compared to previous techniques [31]. ISSR markers allowed effective genetic differentiation between hop varieties without requiring prior knowledge of the DNA sequence examined [31]. However, with advances in next-generation sequencing technologies, new markers systems have emerged: SSR (Simple Sequence Repeated) markers, or microsatellites, and SNP (Single-Nucleotide Polymorphism). A significant number of polymorphic microsatellite [32–36] and SNP [37–40] loci have since been identified and characterized. Moreover, with the availability of whole-genome sequencing data, microsatellites have evolved to EST-SSR (Expressed Sequence Tag – Simple Sequence Repeated)

markers, enabling the targeting of loci located within or near specific genes, thereby providing insights into functional genetic variation. This approach has already demonstrated its effectiveness in assessing genetic diversity among hop cultivars and wild hops [41,42]. This type of markers has been more effective than SNP markers [43]. Many large-scale studies using such strategies have been conducted, for example, at the continental scale [16] or the country scale [44]. However, few studies have been reported on a regional scale.

In this study, we focused on the genetic and metabolic diversity of 36 wild hops (*H. lupulus* var. *lupulus*) collected in northeastern France. To investigate their characteristics rather than directly studying the *in situ* metabolic content of flowers, we pursued an alternative approach. In spring, we collected stems from these wild hops, performed aeroponics cuttings, and cultivated each individual under the same environmental conditions until harvest at the end of the summer. By adopting this approach, we aimed to evaluate and isolate the impact of the hop genotype on its chemical phenotype. First, we assessed the metabolic content in the leaves through targeted metabolite profiling, focusing on compounds with potential applications in the brewing industry and beyond. Furthermore, we used sixteen EST-SSR microsatellites [45,46] to study the genetic proximity of these wild hops. Thus, our research focused on three main areas: metabolic diversity, genetic diversity, and metabolic content prediction. Specifically, we hypothesize that (i) hop leaves may provide a means to distinguish wild hops on the basis of their metabolic profiles, (ii) wild hop collection results in metabolic diversity, (iii) genetic diversity, and (iv) microsatellite analysis can serve as a predictive tool for metabolic production in wild hops.

## Materials and methods

### Material sampling and preparation of wild hop samples

Based on field investigations, floristic databases, and surveys of brewers at the regional scale (range of 30 km from Nancy), exclusively wild hops were randomly collected in various ecological habitats (forest edges, hedges, riparian zones, and field margins) during spring 2020, following their natural distribution. No cultivated hop varieties were included in the sampling. A total of thirty-six wild hop vegetative parts were sampled (12 males and 24 females), and aeroponic cuttings were prepared from the main stem collected. The cuttings were cultured for 3 weeks under aeroponic conditions to develop roots before each individual was transferred to potting soil. No pesticides or chemical fertilizers were applied to the hops during their development cycle. Finally, this collection was placed in a dedicated outdoor space before harvest.

### Metabolomic analysis

All chemical solutions (methanol, acetonitrile, and formic acid) were obtained from the same supplier (Carlo Erba Reagents S.A.S., Val-de-Reuil, France).

**Harvest of wild hop samples.** Wild hops were grown for 6 months under uniform conditions. The vegetative parts were harvested and stored at -20 °C before being freeze-dried for 2 days (Christ Alpha 1–4 LD plus, Grosseron, France).

**Sample preparation and extraction of specialized metabolites from hops.** After freeze-drying, the leaves were collected and ground via a planetary ball mill (Pulverisette 6, Fritsch, Germany), resulting in a powder that was sieved through a 200 μm sieve (Saulas, Grosseron, France). A double maceration extraction was performed on 100 mg (± 0.2 mg) of lyophilized powder with 800 μL of MeOH 80% (MeOH/$H_2O$ 80/20 (v/v)) at room temperature (3 technical replicates/sample). The sample was homogenized via a vortex agitator for 10 min at maximum speed (Vortex Genie 2, Scientific Industries, New York, USA), sonicated at 37 kHz for 10 min in an ultrasonic bath (in "sweep" mode) (Elmasonic S 70, Singen, Germany), and finally centrifuged for 10 min at 4 °C at 9840 × g. Then, 400 μL of the supernatant was collected, and a second extraction was performed on the pellet under the same conditions, except that 600 μL of the supernatant was taken and homogenized with the 400 μL collected previously. Finally, 800 μL of the total supernatant was dried in a speed vacuum device for 14 h at room temperature (Eppendorf Concentrator Plus, Hamburg, Germany).

**Characterization of specialized metabolites from dry extracts.** Fifteen milligrams of dry extract was solubilized at a 1/20 ratio (m/v) in 80% MeOH (MeOH/$H_2$O (pure) 80/20 (v/v)) at room temperature following the same homogenization, sonication, and centrifugation conditions described above. The total extracts were filtered through a 0.2 μm filter (Minisart® RC 4, Sartorius, United Kingdom). Finally, 99.4 μL of these extracts were mixed with 0.6 μL of epoxyaurapten (15.9 mM) (Extrasynthese, Gernay, France), which was used as an internal standard.

**UHPLC-ESI-MS analysis.** Chromatographic analyses were performed on a Vanquish UHPLC system equipped with a binary pump, an autosampler, and a temperature-controlled column. Metabolites contained in the extracts (10 μL injected) were separated on a ZORBAX Eclipse Plus C18 (95 Å, 100 × 2.1 mm, 1.8 μm; Agilent Technologies, Waldbronn, Germany) using a mobile-phase gradient as explained previously [47]. HRMS[1] detection was performed on an Orbitrap IDX™ (Thermo Fisher Scientific, Bremen, Germany) mass spectrometer in positive and negative electrospray ionization (ESI) modes. The capillary voltages were set at 3.5 kV and 2.35 kV for the positive and negative modes, respectively. The source gases were set (in arbitrary units $min^{-1}$) to 40 (sheath gas), 8 (auxiliary gas), and 1 (sweep gas), and the vaporizer temperature was 320 °C. Full-scan MS spectra were acquired from 120 to 1200 *m/z* at a resolution of 60,000. MS[2] analysis was performed for quality control (QC = mix of all extracts) via the AcquireX data acquisition workflow developed by Thermo Fisher. In brief, this workflow increases the number of MS[2] acquisitions, especially for low-intensity ions, through the creation of an inclusion list after the first injection of the sample and the establishment of a dynamic exclusion list via iterative sample analysis (involving 5–6 successive injections). Analytical sequence was conducted, incorporating blank samples (representing the extraction solvent) and quality control (QC) samples inserted at regular intervals between the extracts.

**Molecular identification and quantification.** To profile the metabolites in our different hop extracts, the raw data files were uploaded into Compound Discoverer™ software (version 3.3) (Thermo Fisher Scientific, Bremen, Germany). Briefly, the software workflow included peak detection, chromatogram alignment, and peak grouping in features and raw data files based on blank, QC, and sample files. Each feature corresponds to a specific *m/z* at a given retention time. The compounds were identified through (i) elemental composition prediction; (ii) searching in mass/formula databases (including internal databases (see below) and 9 public databases (AraCyc, BioCyc, Carotenoids Database, FooDB, Golm Metabolome Database, Human Metabolome Database, KEGG, LipidMAPS, and Sequoia Research Products)); and (iii) with MS[2] information, searching in-house and the public spectral databases mzCloud, Mona, and GNPS. When possible, the identification of metabolites was confirmed by comparison with authentic commercial standards acquired via MS[1] and MS[2] under the same chromatographic conditions: hulupinic acid (Merck KGaA, Darmstadt, Germany), α- and β-acids (n, co-n, ad-n) (ICE-4, kindly provided by Régis Boulogne, Kronenbourg), xanthohumol, and isoxanthohumol (Hopsteiner, Mainburg, Germany). Quantification was performed on the molecules found in the extracts for which we had commercial standards, including bitter acids (α and β acids (n, co-n, ad-n)), xanthohumol, and hulupinic acid in the hop leaf extracts. Stock solutions of xanthohumol, ICE-4, and hulupinic acid standards were prepared at a concentration of 1 mg/mL in MeOH for quantification. Ten working solutions ranging from 0.5–650 ng/μL (0.5–5–15–25–50–100–200–350–500–650) were prepared by mixing each standard with an internal standard as previously described. The quantification method was developed via XCalibur software (via Processing Setup and Quan Browser applications; Thermo Fisher Scientific). Briefly, this method involved the automatic integration of the peaks of each molecule by performing a mass (*m/z*) and retention time search (Processing Setup). Finally, this quantification method was tested across the different concentrations studied and applied to our samples to quantify the targeted compounds (Quan Browser).

## Statistical analyses

Statistical analyses were performed with peak area data recovered by Compound Discoverer software for the main compounds in RStudio software (version 4.2.2) [48] via the packages 'FactoMineR' [49], 'factoextra' [50], 'corrplot' [51],

'Polychrome' [52] (for PCA analyses) + 'dendextend' [53], and 'gplots' [54] (for HAC analysis) + 'RColorBrewer' [55] (for heatmap analysis). Briefly, a PCA was performed to explore metabolic data to deduce the variability explained by the variables (targeted metabolites), on which axes to project data, the metabolites contributing the most to the differentiation of samples, and the spatial distribution of the samples. Hierarchical ascending classification (HAC) was tested on all possible combinations of distance calculation methods, as well as the different agglomeration methods, allowing the 2 working methods to yield the most accurate results compared with PCA. Moreover, a heatmap was generated to understand biologically why the samples are distributed in different ways.

### Genetic analysis.

**gDNA extraction.** Young, healthy, and fresh leaves were harvested from the wild hop collection. Genomic DNA (gDNA) extraction was performed with the E.Z.N.A.® Plant DNA Kit according to the manufacturer's instructions (Omega bio-tek, USA). The DNA quantity and quality were subsequently checked via a spectrophotometer (BioPhotometer, Eppendorg AG, Hamburg, Germany).

**Microsatellite amplification and data processing.** gDNA was used to study the genetic diversity of wild hops via 16 microsatellite marker loci employing primers labeled with a fluorescent dye (Generi Biotech, Hradec Kralove, Czech Republic). This method is widely used to genotype both cultivars and wild hops, and is derived from previous studies [41,42,45,46]. Each PCR was performed in a 25 µL reaction volume including 1 ng/µL final DNA concentration, 12.5 µL of Taq PCR Master mix, 3.5 µL of pure $H_2O$ and 2 µL of each primer (10 nM). The PCR conditions were as follows: initial denaturation at 95 °C for 3 min; 35 cycles of 95 °C for 30 sec, 54 °C for 60 sec, and 72 °C for 90 sec; and a final elongation at 72 °C for 10 min. The PCR products were pooled into 4 panels as described (S1 Table). For each wild hop, Panel 1 consisted of 4/4/4/5 µL of CHSH1/LAR1/SAUR1/WRKY1 loci, respectively; Panel 2 consisted of 4/4/4/4 µL of CEL1/CMPS/CaEFh/MYB12 loci, respectively; Panel 3 consisted of 10/4/4 µL of F3H/GA2/NDBP loci, respectively; and Panel 4 consisted of 8/4/8/6/6 µL of ABI51/ARP1/MYB5/SPL9/ZFP8 loci, respectively. The panels were diluted by adding 40 µL of pure $H_2O$. Finally, 2.5 µL of each panel was mixed with 10 µL of formamide Hi-Di™ (Applied Biosystems, Lincoln, CA, USA) before the last denaturation step at 94 °C for 15 min. The samples were analyzed via capillary electrophoresis in a 3130 Genetic Analyzer Refurb (Applied Biosystems Hitachi High-Technologies Corporation, Tokyo, Japan), and the fluorescent peaks were observed via GeneMapper software (version 5.0) (Applied Biosystems, Lincoln, CA, USA). The observed alleles for each locus were coded as "1" (observed) or "0" (not observed) in a table (binary data matrix) before being subjected to statistical analysis.

### Genetic characterization

For each EST-SSR marker, various standard indices have been calculated, including the number of alleles ($N_A$), expected heterozygosity ($H_e$), observed heterozygosity ($H_o$), and polymorphism information content (PIC). For this purpose, the previously binary data matrix obtained was modified to another matrix by only filling the amplified alleles to have 2 columns per marker, each composed of one amplified allele. The results were obtained by submitting this new data matrix to Cervus software (version 3.0.7) [56].

**Visualization of genetic data via a phylogenetic tree.** To assess the phylogeny between the wild hop collection and worldwide hops (variety and wild), the binary matrix obtained previously (listing the presence or absence of alleles) was subjected to DARwin software (version 6.0.021) [57] combined with a genetic database of worldwide hop varieties and wild hops (data on worldwide hop varieties and wild hops were sourced from the Josef Patzak database). Briefly, from this data matrix, the "Dissimilarity" function (using 1000 bootstraps and Jaccard's coefficient) was used to create a phylogenetic tree via the "Unweighted Neighbor-Joining" method (using 1000 bootstraps). The tree was then exported in "Paup/Nexus" format before being visualized and annotated on the iTOL website (Interactive Tree Of Life, version 6) [58].

**Linking genetic variation to metabolic profiles.** The variability in the quantities of the twelve specialized metabolites among the samples was linked to genetic diversity through the fitting of a multivariate regression tree (MRT). An MRT is a binary tree that formalizes a statistical model $Y = f(X)$, where Y is a vector of dependent variables, and X is a set of independent variables that aim at explaining the variability in Y. Here, Y is the set of quantities of the twelve specialized metabolites, and X is the set of binary variables describing the presence/absence of the different alleles. Prior to analysis, the metabolic data were standardized via the "scale" function in RStudio, and the genetic binary matrix was reformatted by converting "0" and "1" to "FALSE" and "TRUE", respectively. The two datasets were then merged (see S3 Table), and the MRT was constructed via the "mvpart" package (version 1.6–2). The building of an MRT consists of recursively splitting data into two nodes such that the intranode variance of Y is minimized (or equivalently, the internode variance is maximized) [59]. Starting from a single root node (the whole dataset), this recursive process generates a binary tree structure in which the terminal nodes are called leaves. Each split is defined as a threshold on an X variable, i.e., the presence or absence of a given allele. A terminal node gathers a subset of samples that are similar with respect to metabolic variability (Y values) and is characterized by the presence or absence of alleles (X variables) that appear on the path from the root node. In our study, the MRT generated was performed with five groups.

## Results

### Analysis of the metabolic composition of wild hops

**Specialized metabolite composition of hop leaves.** To ensure that differences in metabolic profiles reflect only genetic variability, wild hops were cultivated under uniform environmental and soil conditions. Our initial hypothesis suggested that hop leaves may provide a means to distinguish wild hops based on their metabolic profiles. To test this, analyses were conducted using data acquired in negative ionization mode, leading to the identification of twelve metabolites in the extracts: cohulupone, hulupinic acid, hulupone, adhulupone, xanthohumol, cohumulone, humulone + adhumulone, desoxyhumulone, postlupulone, lupulone E, colupulone and lupulone + adlupulone. The main molecules identified were α-type bitter acids (co, n, ad, humulone) and β-type bitter acids (co, n, ad, lupulone). The β-acids were much more common than α-acids. In addition to bitter acids, oxidized derivatives of β-acids, namely, hulupinic acid, as well as co, n- and ad-hulupone, were highlighted. Moreover, we also identified xanthohumol, a well-known flavonoid in hops. Finally, other molecules derived from bitter acids, such as postlupulone, lupulone E (β-acids), and desoxyhumulone (α-acids), were identified (S1 Fig and S2 Table).

### Assessment of metabolic diversity on the basis of the metabolic content of hop leaves.

To assess the metabolic diversity among the wild hop leaves, we subjected the data to multivariate analysis. The PCA biplot (Fig 1) clearly revealed metabolic differences among the wild hop samples, highlighting their distinct metabolic profiles. Considering the twelve targeted specialized metabolites, we explained 76% of the observed variance (PC1 53.7%, PC2 22.3%). The observed variability in the metabolic content of hops cannot be attributed to the sex of the samples, as their points are clustered near the center of the PCA. Moreover, the most contributing specialized metabolites explaining the greatest difference between wild hops are bitter acids (α- and β-type) and β-type oxidized derivatives. At first glance, our collection seems to be divided into 3 distinct groups, one of which is formed only by sample #3, which appears to be a separate individual.

This tendency was confirmed by an unsupervised hierarchical ascending classification (HAC), in which a clear separation into 3 groups was observed (S2 Fig). For the PCA-biplot analysis, sample #3 is separate from the other samples (see the blue branch in S2 Fig). The samples located on the left side of the PCA biplot correspond to the red cluster, and the green cluster gathers all the other samples. We can also observe in the HAC that most replicates per sample are

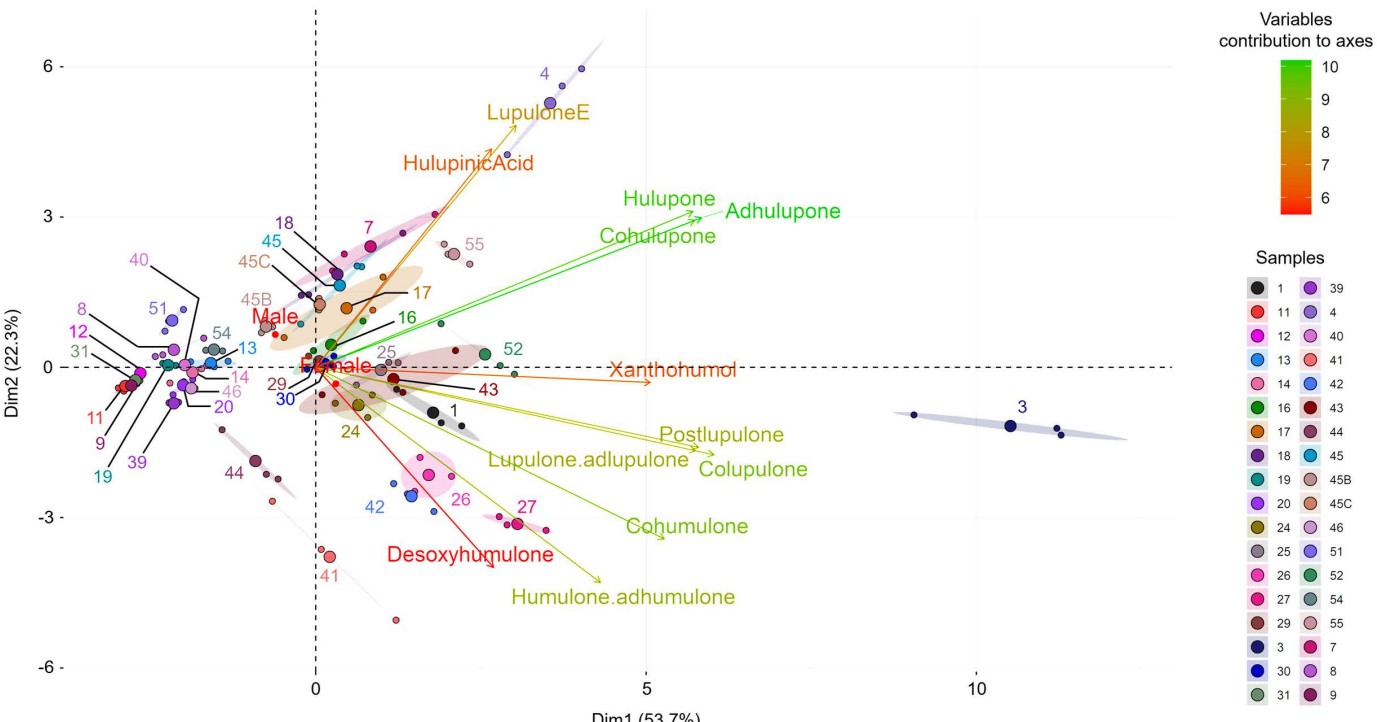

**Fig 1. Principal component analysis (PCA) biplots based on hop leaf content among our wild hop samples (36 samples).** The variable contributions to the axes are shown in the gradient from red (low) to green (high).

clustered together, or at least close to each other, with the exception of a few samples [7,17,43] (S2 Fig), whose ellipses appear more elongated than the others (Fig 1). To understand the origin of these differences, a heatmap was generated (Fig 2). This allowed us to show that cluster 1 (red) was represented by the samples that produced fewer specialized metabolites (corresponding to the left side of the PCA). Cluster 2 (green) is composed of two subgroups; the first is enriched in oxidized β-acid derivatives, and the second is rich in bitter acids (present in the middle of the PCA). Finally, cluster 3 (blue), formed by only one individual (sample #3, on the right side of the PCA), appears to be the cluster with the highest content of specialized metabolites studied. The difference between the different samples seems to be clearly established in terms of the quantity of specialized metabolites produced.

The quantification of molecules with available commercial standards was performed, allowing us to more clearly distinguish differences in quantity among the three clusters (Table 1).

Significant variations were observed between these clusters for β-acids (colupulone, lupulone, and adlupulone) and, to a lesser extent, for α-acids (cohumulone, humulone, and adhumulone), and xanthohumol. For these different metabolites, the highest concentration was observed in cluster 3, followed by cluster 2 and then cluster 1. In contrast, hulupinic acid did not exhibit notable concentration differences between the clusters. These findings further support the initial outcomes from PCA and heatmap analyses: the observed variation between clusters lies in the amount of metabolites produced.

## Analysis of genetic diversity and population structure

**Genetic diversity among wild hops.** To investigate the genetic diversity of the wild hop collection, we used sixteen molecular markers. A total of 76 alleles were detected among the 36 wild hop genotypes, with an average of 4.80 (± 1.57) alleles per locus and a mean PIC of 0.56 (± 0.17) (Table 2).

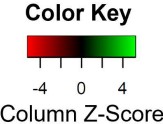

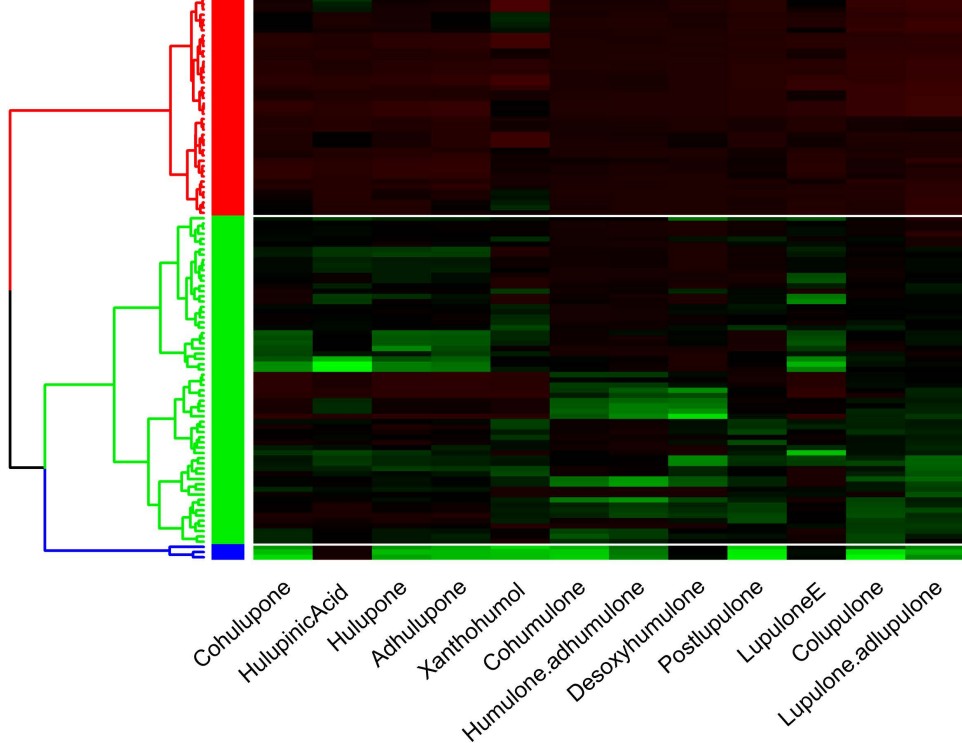

**Fig 2. Heatmap of the metabolic content of hop leaves from the 36 wild hops according to the three clusters determined in the HAC.** A dendrogram was constructed using the "Manhattan" distance and "ward.D2" clustering method.

**Table 1. Quantification of hulupinic acid, xanthohumol, and bitter acids according to the three metabolic clusters. x represents the peak area found in the samples belonging to the cluster. The values (ng/µL) correspond to the concentration point of the tested range, adjusted according to the purity of each standard used.**

| | Compounds | | | | | |
|---|---|---|---|---|---|---|
| | Hulupinic acid | Xanthohumol | Cohumulone | Humulone + adhumulone | Colupulone | Lupulone + adlupulone |
| Cluster 1 (red) | $x<0,5$ | $0,38>x<3,75$ | $x<0,05$ | $x<0,16$ | $0,07>x<1,95$ | $0,07>x<3,38$ |
| Cluster 2 (green) | $0,5>x<5$ | $0,38>x<3,75$ | $0,05>x<0,55$ | $0,16>x<1,58$ | $0,65<x<13,02$ | $0,65<x<27,04$ |
| Cluster 3 (blue) | $x<0,5$ | $3,75<x<11,25$ | $0,55<x<1,65$ | $1,58<x<4,74$ | $45,57<x<65,1$ | $47,32<x<67,6$ |

According to these results, three phylogenetic trees were constructed, the first one based on samples only, the second one based on samples and worldwide varieties, and the last one based on samples, worldwide varieties, and worldwide wild hops genotyped to date. In all the cases, the three major phyla are represented in blue, gray, and yellow (Fig 3). According to these different trees, the results revealed significant genetic diversity among the wild hop samples, and the

**Table 2. Characteristics of the sixteen expressed sequence tag-simple sequence repeat (EST-SSR) loci used to assess the genetic diversity of the wild hop collection.**

| Panel n° | Loci | $N_A$ | $H_O$ | $H_E$ | PIC |
|---|---|---|---|---|---|
| 1 | CHSH1 | 6 | 0.639 | 0.682 | 0.626 |
|  | LAR1 | 8 | 0.806 | 0.791 | 0.75 |
|  | SAUR1 | 2 | 0.333 | 0.351 | 0.286 |
|  | WRKY1 | 4 | 0.667 | 0.542 | 0.474 |
| 2 | CEL1 | 4 | 0.583 | 0.624 | 0.551 |
|  | CMPS | 4 | 0.556 | 0.599 | 0.542 |
|  | CAEFh | 4 | 0.5 | 0.511 | 0.457 |
|  | MYB12 | 6 | 0.694 | 0.714 | 0.654 |
| 3 | F3H | 7 | 0.472 | 0.707 | 0.656 |
|  | GA2 | 4 | 0.72 | 0.7 | 0.63 |
|  | NDBP | 5 | 0.583 | 0.763 | 0.71 |
| 4 | ABI51 | 6 | 0.806 | 0.82 | 0.782 |
|  | APR1 | 3 | 0.222 | 0.205 | 0.19 |
|  | MYB5 | 3 | 0.36 | 0.41 | 0.36 |
|  | SPL9 | 5 | 0.778 | 0.746 | 0.699 |
|  | ZPF8 | 5 | 0.778 | 0.716 | 0.652 |
| Mean |  | 4.75 | 0.59 | 0.62 | 0.56 |
| (SD) |  | 1.57 | 0.18 | 0.17 | 0.17 |

samples were grouped into 3 different clades. However, this analysis revealed that, at a larger scale, including all hop varieties genotyped to date (Fig 4), the wild hops are part of the same phylum (in gray).

When including the database of genotyped hop varieties with our samples, we observe that the wild hops collected from the northeastern part of France are primarily clustered together. However, some samples, such as samples 1, 11, 12, 9, 7, 39 and 40, appear genetically distant from the others. Additionally, we note that some samples show genetic affiliation with cultivars grown in our region, such as Strisselspalt, Précoce/Tardif of Burgundy, and Aramis.

Finally, when we included genotyped wild hops (see pink text in Fig 5) originally collected in Europe and North America, our analysis revealed that the majority of the wild hops collected belonged to the same phylum (in gray), with the exception of two samples [8 and 13], which appeared in the yellow phylum.

Our collected wild hops are always affiliated with typical European varieties but appear to be much more genetically distinct with the addition of a database from genotyped wild hops. Strong proximity with the wild hops from a mountainous area in the Czech Republic (see "Jeseniky" in Fig 5) or other European countries, such as Sweden, Switzerland, Spain, and France, can be observed. The wild hops from North America (Canada, USA) are in a distinct clade in the blue phylum. Finally, our different samples, which are colored according to their metabolic cluster (e.g., colors red, blue, and green), do not form distinct clusters in the phylogenetic trees.

**Analysis of genetic determinants of metabolic diversity.** Finally, to assess whether metabolite production in wild hops can be predicted via microsatellite markers, we conducted a multivariate regression tree (MRT) analysis. An MRT can be considered a variance-partitioning tool by considering the ratio between the overall variance in metabolite quantities among the 36 samples and the sum of the residual variance in each terminal node. The fitted MRT has a residual error = 0.604, which means that the model's $R^2$ is 0.396 (1–0.604), so the tree explains 39.6% of the variance in the metabolic matrix (Fig. 6).

Two molecular markers are involved in the metabolic differentiation of hops: WRKY1 (with three alleles size) and CHSH1 (with one allele size). Additionally, the allele with a size of 224 for the WRKY1 marker appears to be a major

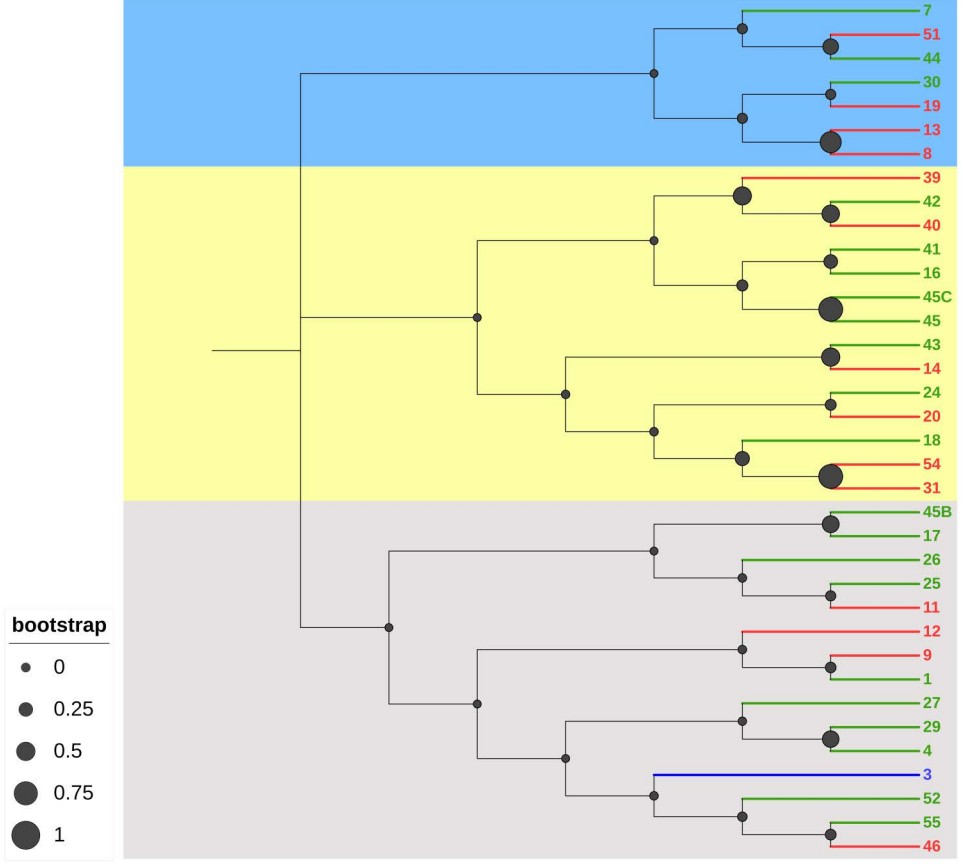

**Fig 3. Phylogenetic tree of 36 wild hops based on molecular data, calculated via the unweighted neighbor-joining method (based on Jaccard's coefficient).** The samples are colored according to their metabolic cluster (cluster 1: red; cluster 2: green; or cluster 3: blue). The filled black circles indicate bootstrap values.

determinant between the different metabolic clusters, as it is present in the differentiation of the first two nodes. However, the MRT failed to correctly predict metabolite quantities of unseen samples (cross-validated (CV) error greater than 1, S3 Fig); therefore, the link between metabolic and genetic variability cannot be generalized to other datasets and is presented here for a purely descriptive purpose.

## Discussion

In this study, we explored the genetic and metabolic diversity of *H. lupulus* (L.) wild germplasm from the northeast region of France.

To assess genetic diversity, we adopted a fingerprinting approach using 16 EST-SSR markers [41,42]. This method has already been used and is reliable for establishing genetic links between cultivars [45,46]. These molecular markers led us to analyze allele amplification, the number of alleles per locus, the PIC values, and the heterozygosity rates of our wild hop collection. Compared with other studies, we amplified fewer alleles per locus, resulting in a lower mean PIC [20,22,27,60]. However, previous studies suggest that PIC values higher than 0.5 can be considered informative [61]. Regarding the average level of heterozygosity, we found that the observed heterozygosity ($H_O$) was lower than the expected heterozygosity ($H_E$): 0.59 and 0.62, respectively (Table 2). This result suggests a slight heterozygote deficit in

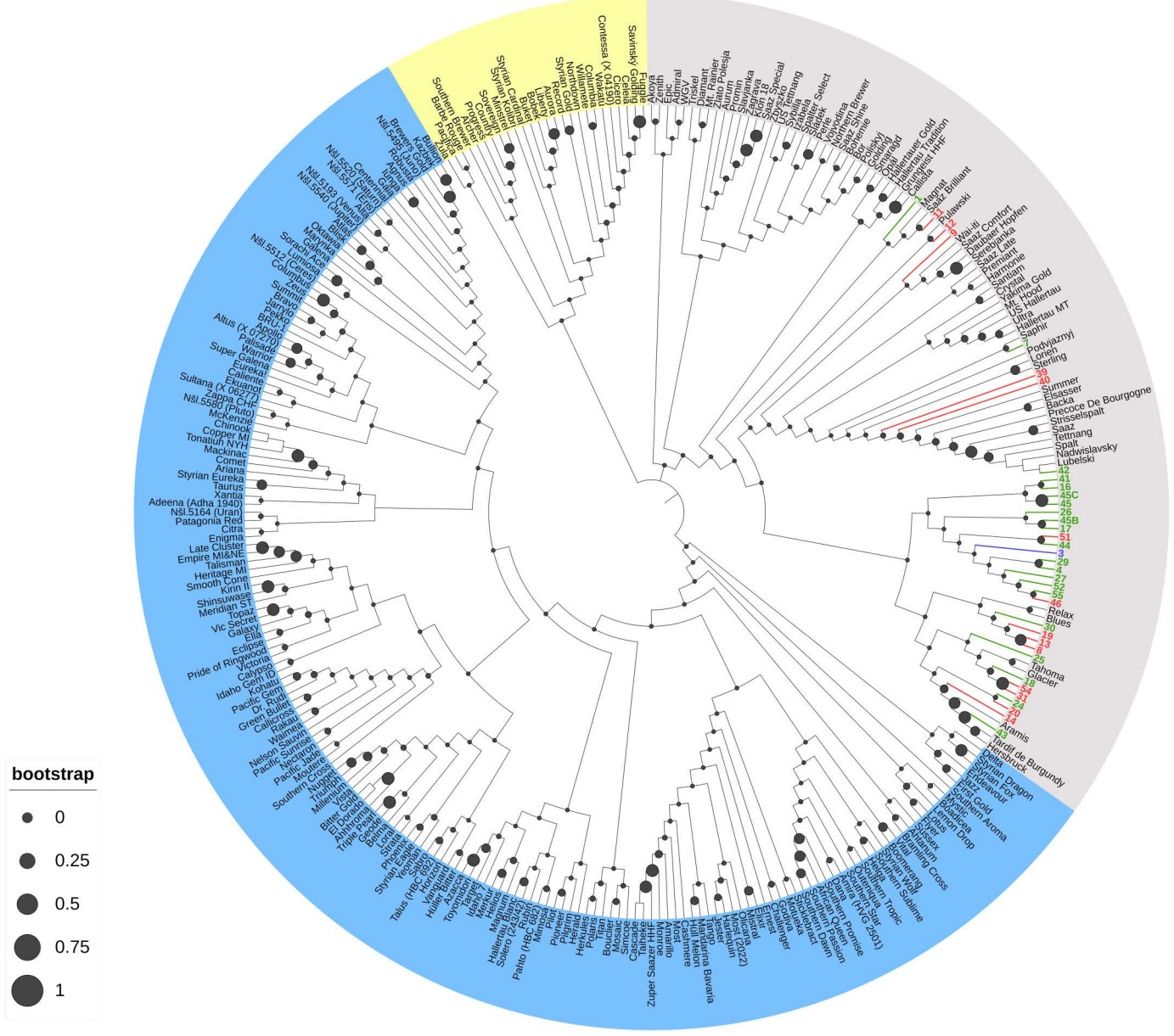

**Fig 4. Phylogenetic trees of 36 wild hops and worldwide hop varieties sampled to date, on the basis of molecular data, calculated via the unweighted neighbor-joining method (based on Jaccard's coefficient).** The samples are colored according to their metabolic cluster (cluster 1: red; cluster 2: green; or cluster 3: blue), and the hop varieties worldwide are black. The filled black circles indicate bootstrap values.

our collection but remains consistent with studies performed in other regions, such as northern France [60]. On the basis of these results, we hypothesize that many of the collected wild hops probably originated from an older cultivar, possibly escaping from hop fields by sexual reproduction and seed dissemination when hop cultivation was greatly reduced or even abandoned during the World Wars. Second, this slight heterozygote deficit may also reflect the impact of environmental or geographical factors limiting gene flow. This situation has already been reported by Murakami regarding European wild hops from the Caucasus region, which are different from the rest of the European wild hops [34]. Furthermore,

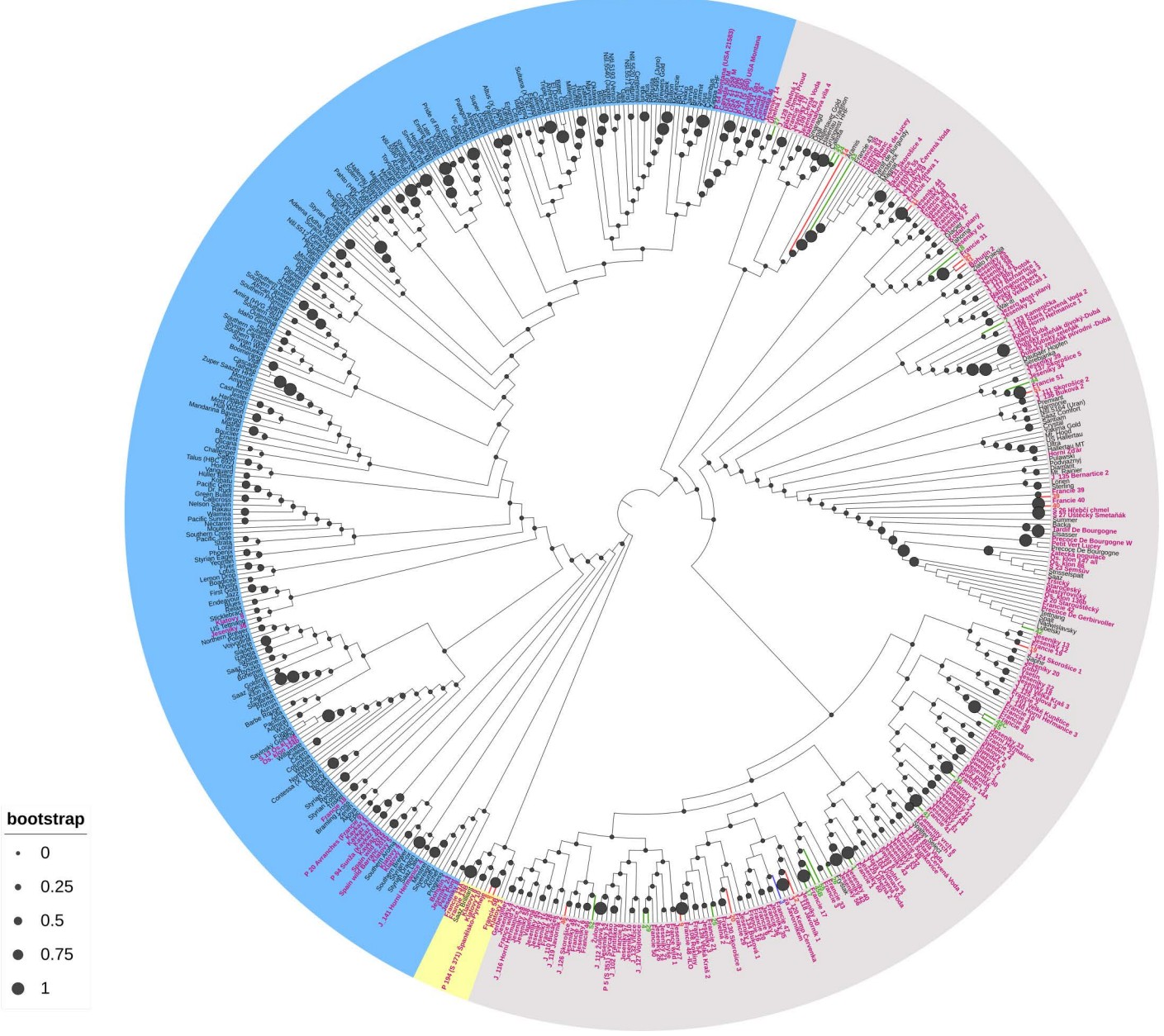

**Fig 5. Phylogenetic trees of 36 wild hops, worldwide hop varieties and wild hops sampled to date, on the basis of molecular data, calculated via the unweighted neighbor–joining method (based on Jaccard's coefficient).** The samples are colored according to their metabolic cluster (cluster 1: red; cluster 2: green; or cluster 3: blue), the worldwide hop varieties are in black, and the worldwide wild hops are in pink. The filled black circles indicate bootstrap values.

our study extends genetic comparisons to worldwide commercial varieties and wild hops from Europe (France, Czech Republic, Switzerland, Sweden, and Spain) and North America (USA and Canada) genotyped to date with this molecular method. Our results confirm the genetic affiliation of our collection with European hops, including wild and cultivated varieties such as Strisselspalt (a variety still cultivated currently in our region), while distinctly separating them from North American hops (Figs 4 and 5). These findings align with the work of Murakami and collaborators, who highlighted the

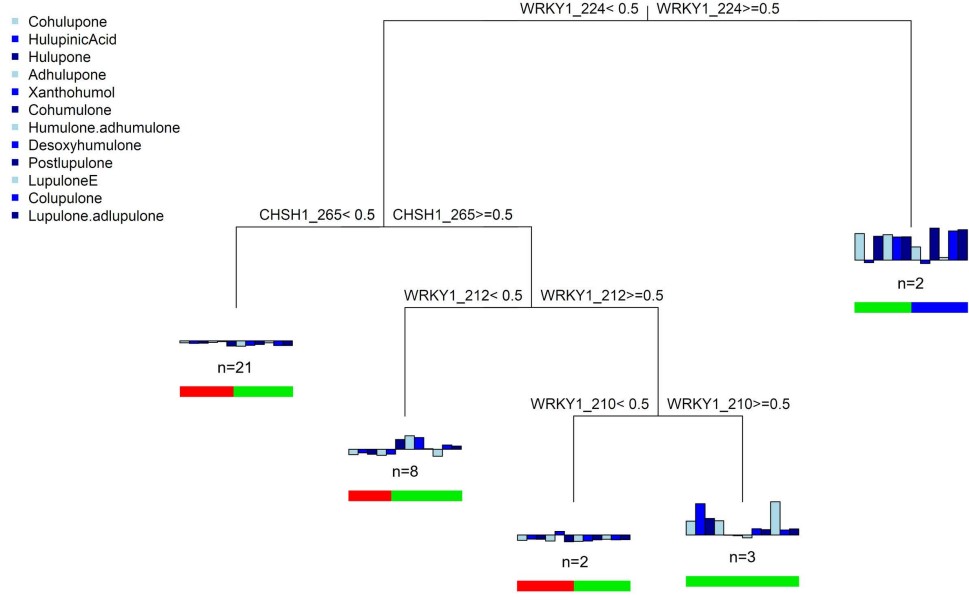

**Fig 6. Multivariate regression tree based on the 36 wild hops genetic and metabolic data.** The MRT is based on five groups. The colored vertical bar plots represent the mean quantity of the twelve metabolites at each node. The colored horizontal bar plots show the proportion of the metabolic cluster represented at each node, according to the colors in Fig 2. Allele presence is labeled ">=", and all other alleles are labeled "<". n represents the number of individuals contained in each node. The statistics at the bottom of the figure are the residual error, the cross-validated error, and the standard error.

divergence between European and North American hops, supporting their hypothesis of a recent and rapid expansion of hops throughout Europe [62]. Moreover, our results are consistent with those of previous studies that identified genetic diversity among European wild hops [16,19,20,24,25,27,44,63].

In addition to the molecular study, we performed a chemotyping approach based on hop leaf composition to assess metabolic diversity in the wild hop collection. Plants are known for their vast production of specialized metabolites, which are often linked to environmental adaptation. In the case of hops, these compounds, especially bitter acids and volatiles, contribute to the brewing industry [64]. To investigate specifically the impact of hop genetics on specialized metabolite production, the collection was grown on the same soil under the same environmental conditions for six months. From hop leaves, we extracted a broad spectrum of soluble compounds and identified typical hop metabolites, such as prenylated chalcone and bitter acids, and their oxidized derivatives, such as hulupinic acid and co-, n-, and ad-hulupone. The finding of these degradation products in our extracts was surprising because only visually unoxidized leaves were retained for analysis. However, it is not impossible for these molecules to appear since they are intermediate in the oxidation of β-acids [5]. This might also be explained by a late harvest (in October) when the degradation of the major compounds could have started [65]. Morover, our analysis revealed that β-acids were more abundant than α-acids, a characteristic already observed with wild hops collected in northern France [60], which contrasts with commercial varieties known for higher α-acid yields. This observation further supports the distinct "wild" nature of our collection.

In hop cultivation, female flowers are harvested for use in the brewing industry, whereas vegetative parts (leaves and stems) are often left as field biomass [3,66]. Recent studies suggest that leaves could have additional valorization potential. For example, Velot and collaborators reported that hop leaf extract from the Cascade variety has anti-inflammatory properties *in vitro* when it is applied to human bone cells [47]. This finding is economically relevant given that leaves and

stems can constitute up to 75% of the dry weight of a mature hop plant [67,68]. In light of these findings, with wild hops leaves showing concentrations of up to 68 ng/μL for β-acids and 11 ng/μL for xanthohumol, exploring the potential for hop leaves as a valuable resource, similar to hop flowers within the brewing industry, would be worthwhile.

Finally, in our study, we evaluated the genetic influence on metabolic production, a topic that was previously attempted in the literature without success [16,17]. To perform this analysis, we set up a simple, quick, and efficient method to propagate hops based on aeroponic culture, followed by a cultivation step under uniform conditions. In our study, the MRT analysis created five groups (Fig 6) highlighted two molecular markers, WRKY1 (with three allele sizes) and CHSH1 (with one allele size), related to the content of specialized metabolites. These results are consistent with other studies that revealed the involvement of WRKY1 in the activation of essential genes in the biosynthetic pathway of prenylated flavonoids such as xanthohumol and of bitter acids [69,70]. Other studies have also shown the role of CHSH1 in the biosynthesis of flavonoids, as well as bitter acid [71,72]. On the basis of these previous results, we can hypothesize that the metabolic differences observed among the three clusters are attributable to genetic variations in regions implicated in biosynthetic pathways, which may lead to changes in metabolic regulation. On the other hand, the observed metabolic differences may be attributed to phenotypic differences. Indeed, it has been demonstrated that the number and size of lupulin glands affect the contents of polyphenols and bitter acids [73]. Moreover, Morcol et al. (2021) reported that the leaves of *Humulus neomexicanus*, which have high contents of bitter acids or prenylflavonoids, had significantly greater density of glandular trichomes [74]. Unfortunately, this aspect was not addressed in our study. Finally, on the basis of the MRT results, our analysis did not demonstrate predictive power (CVRE value > 1). It would therefore be interesting to perform the same genetic and metabolic studies, with the same experimental design (i.e., on the same type of soil and under the same environmental conditions), on wild hops and high-alpha and aroma hop varieties to confirm, or not, the trends observed in the MRT analysis on a larger sample size.

## Conclusion

In this study, we analyzed the diversity of 36 plants collected over a geographical area of 30 km² in northeastern France. These analyses revealed that the use of a molecular marker-based approach is a robust way to study genetic closeness with worldwide cultivars and wild hops. Our results provide evidence that the wild hops collected are characterized by a high degree of genetic variability. Our study also shows that another way to evaluate the diversity of this population might rely on metabolic analyses. However, in this case, the difference is not found in terms of the nature of the molecules produced but rather in terms of quantity. The comparison of genetic and metabolic data allowed us to identify two molecular markers (WRKY1 and CHSH1), which provided effective metabolic discrimination of wild hops on the basis of genetic information. Thus, metabolic differences observed in our study can rely on genetic variations in specific regions implicated in biosynthetic pathways of specialized metabolites studied here.

## Supporting information

**S1 Fig. UHPLC-MS chromatogram ([M-H]⁻) from Quality Control (QC) of hop leaf extract.** Identified peaks are annotated from 1 to 12 according to S2 Table. NL: Normalization Level, FTMS: Fourier Transform Mass Spectrometry, ESI: Electrospray Ionisation. (XCalibur software - Qual Browser application, Thermo Fisher Scientific).
(DOCX)

**S2 Fig. Hierarchical Ascending Classification (HAC) analysis based on hops leaf metabolic content.** The resulting dendrogram was made using the "Manhattan" distance, and "ward.D2" clustering method.
(DOCX)

**S3 Fig. Selection of the MRT for the wild hops genetic and metabolic data.** The resulting figure show the relative error (RE) in green, and the cross-validated relative error (CVRE) in blue of trees of increasing size. A CVRE above one indicates that the MRT fails to correctly predict unseen data. The vertical bars indicate one standard error for the CVRE,

and the red line indicates one standard error above the minimum CVRE. Orange dot show the smallest tree within one standard error of the CVRE. Lime green bars indicate the number of times each tree size was chosen during the cross-validation process.
(DOCX)

**S1 Table. Details of the sixteen EST-SSR (expressed sequence tag-simple sequence repeat) loci used in this study.**
(XLSX)

**S2 Table. List of twelve hop metabolites identified in extracts.**
(XLSX)

**S3 Table. Original data for the MRT analysis.**
(XLSX)

## Author contributions

**Conceptualization:** Séverine Piutti, Alexandre Laflotte, Alain Hehn.

**Data curation:** Josef Patzak, Alain Hehn.

**Formal analysis:** Florent Ducrocq.

**Funding acquisition:** Florent Ducrocq, Séverine Piutti, Alain Hehn.

**Investigation:** Florent Ducrocq, Alena Henychová.

**Methodology:** Florent Ducrocq, Alexandre Laflotte, Loïc Girardeau, Jérémy Grosjean.

**Project administration:** Séverine Piutti, Alain Hehn.

**Resources:** Florent Ducrocq, Séverine Piutti, Alain Hehn.

**Software:** Florent Ducrocq, Alena Henychová, Jean Villerd.

**Supervision:** Séverine Piutti, Josef Patzak, Alain Hehn.

**Validation:** Séverine Piutti, Josef Patzak, Alain Hehn.

**Visualization:** Florent Ducrocq.

**Writing – original draft:** Florent Ducrocq.

**Writing – review & editing:** Séverine Piutti, Jean Villerd, Alain Hehn.

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
