## [Decision Letter · Decision Letter 0]

13 Feb 2025

PONE-D-24-56047Fingerprinting and chemotyping approaches reveal a wide genetic and metabolic diversity among wild hops (*Humulus lupulus* L.).PLOS ONE

Dear Dr. Hehn,

Thank you for submitting your manuscript to PLOS ONE. After careful consideration, we feel that it has merit but does not fully meet PLOS ONE’s publication criteria as it currently stands. Therefore, we invite you to submit a revised version of the manuscript that addresses the points raised during the review process.

We look forward to receiving your revised manuscript.

Kind regards,

Suman S. Thakur, Ph.D

Academic Editor

PLOS ONE

Journal Requirements:

**Additional Editor Comments:**

Major Revision

Reviewers' comments:

Reviewer's Responses to Questions

**Comments to the Author**

1. Is the manuscript technically sound, and do the data support the conclusions?

Reviewer #1: No

Reviewer #2: Yes

2. Has the statistical analysis been performed appropriately and rigorously? 

Reviewer #1: No

Reviewer #2: Yes

3. Have the authors made all data underlying the findings in their manuscript fully available?

Reviewer #1: Yes

Reviewer #2: Yes

4. Is the manuscript presented in an intelligible fashion and written in standard English?

Reviewer #1: Yes

Reviewer #2: Yes

5. Review Comments to the Author

Reviewer #1: The authors investigated the genetic and metabolic diversity of 36 varieties of hops in France and evaluated the molecular characterization using 16 SSR markers. Finally, they highlighted that WRKY 224 was molecular determinates for metabolic production. However, this study is a preliminary analysis of genetic diversity of hops, lacking in-depth analysis. The two major concerns are the sample selection and gene identification, which need more detail biological evidence. Here are some comments.

1.Introduction: This section needs to be more scientificity, shortening the history of hops and focusing on the genetic diversity and molecular marker analysis of wild varieties of hops. Since the review of molecular markers development is too general, which make it difficult to get the recent progress.

2.Materials and Methods: The samples selected in this paper looks randomly, and the reason why you choose these materials should be charify. Are these materials representing the natural distribution, just a regional distribution, or varieties widely used in production? All of these will affect the results of this study.

Line 317: How did you choose these 16 EST-SSR markers?

Line 345: This results should be analyzed further and more data should be provided in this section. For example, the original data of MRT should be provided. Why did the residual error of 0.606 represent the R2 of 39.6%? There was a statistical gap in this section. Additionally, how many genes involved in the metabolic differentations, and the reasons why you choose the two genes lacked the genetic evidence.

Line 453: Why did the authors point out the light conditions?

Reviewer #2: The present manuscript builds on the hypothesis that hop leaves may provide a means to distinguish wild hops based on their specialized metabolic profile. The selected metabolites include polyketide-derived prenylated phenolics, such as alpha and beta acids (hulupone and lupulone derivatives, respectively), which contribute to bitterness, and a prenylated flavonoid (a humol derivative), which contributes to antioxidant properties. The study is based on the metabolic and genetic examination of 36 wild hop accessions collected in the vicinity of Nancy, France, from various ecological habitats, such as forest edges, hedges, riparian zones, and field margins. The samples were grown and propagated under uniform conditions to enable a direct comparison of their chemical profiles. In addition, genomic DNA was analyzed using 16 hop-specific microsatellite marker loci to study genetic diversity and classify the accessions relative to global genetic data.

The metabolic profiles varied in intensity across twelve targeted metabolic identifiers: cohulupone, hulupinic acid, hulupone, adhulupone, xanthohumol, cohumulone, humulone + adhumulone, desoxyhumulone, postlupulone, lupulone E, colupulone, and lupulone + adlupulone. Although the authors conclude that the accessions group into three metabolic profiles, the data suggest that these profiles are distinguished by intensity rather than by compositional differences. Cluster analyses suggest that some WRKY enzymes could be responsible for this variation, potentially offering insight into metabolic regulation rather than differences in biosynthetic pathways. While this is briefly discussed in the conclusion, it could benefit from greater emphasis in the main text.

The paper also places the collected hop population into a broader context by incorporating global genetic data on hops, as described in the methods section. As expected, the accessions clustered within the European genetic group, and the authors conclude that the accessions likely originated from a single naturalization event. It would have been interesting if the authors had included hops from other parts of the world as out-groups, which might have provided additional insights into novel aspects of the metabolism of polyketide-derived prenylated phenolics in hops.

The manuscript is well prepared, and the analyses are robust and clearly presented. However, it appears that some of the 36 accessions are not included in every global comparison. It would have been beneficial to see all accessions consistently included in all phylogenetic analyses.

The discussion is highly relevant and well-aligned with the results. If anything, the term “chemical grouping” used to describe the division among accessions in Figures 1 and 2 might be reconsidered. These figures suggest variations in the intensity of highly similar metabolic profiles, with some accessions being low producers, others having elevated intensity, and one (Genotype 3) being a strong general producer of the compounds of interest. Terms such as “metabolic activity” or “intensity groupings” may be more appropriate descriptors.

6. PLOS authors have the option to publish the peer review history of their article (what does this mean? ). If published, this will include your full peer review and any attached files.

**Do you want your identity to be public for this peer review?** For information about this choice, including consent withdrawal, please see our Privacy Policy .

Reviewer #1: No

Reviewer #2: **Yes: ** Benedicte Riber Albrectsen

---

## [Author Response · Author response to Decision Letter 1]

7 Mar 2025

Response to Reviewer

PONE-D-24-56047

Fingerprinting and chemotyping approaches reveal a wide genetic and metabolic diversity among wild hops (Humulus lupulus L.).

Referee: 1

The authors investigated the genetic and metabolic diversity of 36 varieties of hops in France and evaluated the molecular characterization using 16 SSR markers. Finally, they highlighted that WRKY 224 was molecular determinates for metabolic production. However, this study is a preliminary analysis of genetic diversity of hops, lacking in-depth analysis. The two major concerns are the sample selection and gene identification, which need more detail biological evidence. Here are some comments.

Referee’s comment:

Introduction: This section needs to be more scientificity, shortening the history of hops and focusing on the genetic diversity and molecular marker analysis of wild varieties of hops. Since the review of molecular markers development is too general, which make it difficult to get the recent progress.

Author’s answer:

Thank you for this comment. As requested by reviewer 1, we have shortened the history of hops, and developed the part on the genetic diversity and molecular marker analysis in the introduction of the revised manuscript.

Referee’s comment:

Materials and Methods: The samples selected in this paper looks randomly, and the reason why you choose these materials should be charify. Are these materials representing the natural distribution, just a regional distribution, or varieties widely used in production? All of these will affect the results of this study.

Author’s answer:

The 36 samples analyzed in our study are exclusively wild hops, with no cultivated hop varieties included. The 36 wild hops were randomly collected across the region, from various ecological habitats as we mentioned in our manuscript (forest edges, hedges, riparian zones, and field margins). Thus, as you pointed out, they represent a natural distribution since they are wild hops. We have clarified this point in the revised manuscript.

Referee’s comment:

Line 317: How did you choose these 16 EST-SSR markers?

Author’s answer:

We are not certain why this specific line was mentioned, as it does not refer to microsatellites. However, these 16 EST-SSR markers used are the most recent markers developed and used routinely in the lab. This method has proven its effectiveness and is used to genotype hops, both cultivars and wild populations, from around the world. This also explains why we were able to affiliate genetically our samples with various wild hops and cultivars from around the world, as we now have an extensive database. We have specified this in the revised manuscript, and we have added the corresponding references.

Referee’s comment:

Line 345: This results should be analyzed further and more data should be provided in this section. For example, the original data of MRT should be provided. Why did the residual error of 0.606 represent the R2 of 39.6%? There was a statistical gap in this section. Additionally, how many genes involved in the metabolic differentations, and the reasons why you choose the two genes lacked the genetic evidence.

Author’s answer:

To enhance the clarity of our MRT analysis, we have now provided the original MRT data in Supplementary Table S3, as requested by Reviewer 1. A Multivariate Regression Tree (MRT) is a binary tree model that represents a statistical relationship Y = f(X), where Y corresponds to a vector of dependent variables (in our case, metabolic data), and X consists of independent variables (in our case, genetic data) used to explain the variability in Y. In simpler terms, this approach allows us to assess how genetic diversity (X) influences metabolic diversity (Y). Regarding the calculation of R², it is obtained by subtracting the residual error from 1. Based on our results, the residual error is 0.604, leading to an R² of 0.396 (1 – 0.604). This indicates that 39.6% of the variation in the metabolic dataset is explained by the genetic factors involved in this study. Finally, it is important to clarify here that WRKY1 and CHSH1 markers were not preselected for this analysis. Instead, these two markers emerged as key determinants of metabolic diversity after the MRT analysis (three different allele size for WRKY1 and 1 allele size for CHSH1).

Referee’s comment:

Line 453: Why did the authors point out the light conditions?

Author’s answer:

Thank you for that comment. This is a mistake, and this sentence has been removed from the revised manuscript.

Referee: 2

The present manuscript builds on the hypothesis that hop leaves may provide a means to distinguish wild hops based on their specialized metabolic profile. The selected metabolites include polyketide-derived prenylated phenolics, such as alpha and beta acids (hulupone and lupulone derivatives, respectively), which contribute to bitterness, and a prenylated flavonoid (a humol derivative), which contributes to antioxidant properties. The study is based on the metabolic and genetic examination of 36 wild hop accessions collected in the vicinity of Nancy, France, from various ecological habitats, such as forest edges, hedges, riparian zones, and field margins. The samples were grown and propagated under uniform conditions to enable a direct comparison of their chemical profiles. In addition, genomic DNA was analyzed using 16 hop-specific microsatellite marker loci to study genetic diversity and classify the accessions relative to global genetic data.

Referee’s comment:

The metabolic profiles varied in intensity across twelve targeted metabolic identifiers: cohulupone, hulupinic acid, hulupone, adhulupone, xanthohumol, cohumulone, humulone + adhumulone, desoxyhumulone, postlupulone, lupulone E, colupulone, and lupulone + adlupulone. Although the authors conclude that the accessions group into three metabolic profiles, the data suggest that these profiles are distinguished by intensity rather than by compositional differences. Cluster analyses suggest that some WRKY enzymes could be responsible for this variation, potentially offering insight into metabolic regulation rather than differences in biosynthetic pathways. While this is briefly discussed in the conclusion, it could benefit from greater emphasis in the main text.

Author’s answer:

Thank you for this comment. We have completed a sentence in both the main text, and in the conclusion regarding this subject in the revised manuscript. However, our discussion regarding WRKY1 and CHSH1 is based on MRT results where we performed a correlation between hop genetic and metabolic data. As mentioned, previous correlation attempts in literature have been unsuccessful, but MRT analysis offers a novel approach to bridging the gap between genetic and metabolic diversity in hops. Although our findings are consistent with previous studies (specifically on WRKY1 and CHSH1), we have chosen not to develop extensively on this aspect, as the MRT analysis exhibited limited predictive power (CRVE > 1), suggesting that its applicability may be restricted to our datasets (we specified it in the abstract). Therefore, this is why we propose at the end of the discussion that similar analyses should be conducted on larger sample size, for example with bitter/high producer and low-bitterness/low producer varieties, to further confirm the trends observed in the MRT analysis.

Referee’s comment:

The paper also places the collected hop population into a broader context by incorporating global genetic data on hops, as described in the methods section. As expected, the accessions clustered within the European genetic group, and the authors conclude that the accessions likely originated from a single naturalization event. It would have been interesting if the authors had included hops from other parts of the world as out-groups, which might have provided additional insights into novel aspects of the metabolism of polyketide-derived prenylated phenolics in hops.

Author’s answer:

Thank you for this comment. Indeed, adding hops from other origins could have brought complementary elements. However, in our article we chose to stick to local wild hops and see what their genetic and metabolic diversity were. Additionally, while genetic diversity can be relatively easily assessed, metabolic diversity studies under the same experimental conditions as described in our article requires significantly more resources. This process involves finding wild hops in different regions or countries, collecting samples, propagating cuttings, and cultivating them under identical experimental conditions to ensure reliable comparisons. We appreciate your suggestion and acknowledge that future studies incorporating a broader geographic sampling could further expand our understanding of hop metabolic diversity.

Referee’s comment:

The manuscript is well prepared, and the analyses are robust and clearly presented. However, it appears that some of the 36 accessions are not included in every global comparison. It would have been beneficial to see all accessions consistently included in all phylogenetic analyses.

Author’s answer:

The 36 wild hops are included in all figures for phylogenetic (and metabolic) analyses. To clarify this point, we have specified it in all figure’ title in the revised manuscript.

Referee’s comment:

The discussion is highly relevant and well-aligned with the results. If anything, the term “chemical grouping” used to describe the division among accessions in Figures 1 and 2 might be reconsidered. These figures suggest variations in the intensity of highly similar metabolic profiles, with some accessions being low producers, others having elevated intensity, and one (Genotype 3) being a strong general producer of the compounds of interest. Terms such as “metabolic activity” or “intensity groupings” may be more appropriate descriptors.

Author’s answer:

Thank you for this comment. We believe reviewer 2 was referring to the term ‘clusters’ rather than ‘chemical grouping’, as the latter does not appear in the figure titles or the main text. However, we have chosen to retain the term ‘cluster’ because it aligns with the standard terminology used in multivariate analyses conducted in this study (PCA, HAC, heatmap). This term is commonly employed in the literature, regarding metabolomic data analysis to describe groups of samples sharing similarities, such as: Li, D., Heiling, S., Baldwin, I. T., & Gaquerel, E. (2016). Illuminating a plant’s tissue-specific metabolic diversity using computational metabolomics and information theory. Proceedings of the National Academy of Sciences, 113(47), E7610-E7618; Wang, S., Yang, C., Tu, H., Zhou, J., Liu, X., Cheng, Y., ... & Xu, J. (2017). Characterization and metabolic diversity of flavonoids in citrus species. Scientific Reports, 7(1), 10549; Liu, K., Abdullah, A. A., Huang, M., Nishioka, T., Altaf-Ul-Amin, M., & Kanaya, S. (2017). Novel Approach to Classify Plants Based on Metabolite-Content Similarity. BioMed Research International, 2017(1), 5296729; Mochida, K., Furuta, T., Ebana, K., Shinozaki, K., & Kikuchi, J. (2009). Correlation exploration of metabolic and genomic diversity in rice. BMC Genomics, 10, 1-10. Given the consistency of the term ‘cluster’ in the field, we believe it is appropriate to describe the division among wild hops in our study. Nevertheless, we appreciate your suggestion and have ensured clarity in the text regarding the use of this terminology.

---

## [Decision Letter · Decision Letter 1]

19 Mar 2025

Fingerprinting and chemotyping approaches reveal a wide genetic and metabolic diversity among wild hops (*Humulus lupulus* L.).

PONE-D-24-56047R1

Dear Dr. Hehn,

We’re pleased to inform you that your manuscript has been judged scientifically suitable for publication and will be formally accepted for publication once it meets all outstanding technical requirements.

Kind regards,

Suman S. Thakur, Ph.D

Academic Editor

PLOS ONE

Additional Editor Comments (optional):

Reviewers' comments:

Reviewer's Responses to Questions

**Comments to the Author**

1. If the authors have adequately addressed your comments raised in a previous round of review and you feel that this manuscript is now acceptable for publication, you may indicate that here to bypass the “Comments to the Author” section, enter your conflict of interest statement in the “Confidential to Editor” section, and submit your "Accept" recommendation.

Reviewer #2: All comments have been addressed

2. Is the manuscript technically sound, and do the data support the conclusions?

Reviewer #2: Yes

3. Has the statistical analysis been performed appropriately and rigorously? 

Reviewer #2: Yes

4. Have the authors made all data underlying the findings in their manuscript fully available?

Reviewer #2: Yes

5. Is the manuscript presented in an intelligible fashion and written in standard English?

Reviewer #2: Yes

6. Review Comments to the Author

Reviewer #2: I just had a few points to address in my initial review, and my comments on sample sizes have been appropriately addressed. ON line 245, I noticed that the manuscript currently uses ‘These method,’ which seems incorrect. Please revisit this phrase and adjust it to either ‘This method’ or ‘These methods,’ depending on your intended meaning.

7. PLOS authors have the option to publish the peer review history of their article (what does this mean? ). If published, this will include your full peer review and any attached files.

**Do you want your identity to be public for this peer review?** For information about this choice, including consent withdrawal, please see our Privacy Policy .

Reviewer #2: No

---

## [Editor Report · Acceptance letter]

PONE-D-24-56047R1

PLOS ONE

Dear Dr. Hehn,

I'm pleased to inform you that your manuscript has been deemed suitable for publication in PLOS ONE. Congratulations! Your manuscript is now being handed over to our production team.

Kind regards,

on behalf of

Dr. Suman S. Thakur

Academic Editor

PLOS ONE